# Spatial analysis of mortality due to congenital syphilis in Brazil from 2008 to 2022

Yago Tavares Pinheiro[1], Richardson Augusto Rosendo da Silva[1],
Ketyllem Tayanne da Silva Costa[1]*, Angelo Giuseppe Roncalli da Costa Oliveira[1],
Janmilli Dantas da Costa[2], Cristiane da Silva Ramos Marinho[2],
Jurandir Alves de Freitas Filho[3], Luennia Kerlly Alves Rocha[3], Victória Sampaio Moreira[3],
Ruan Carlos de Queiroz Monteiro[3], José Rebberty Rodrigo Holanda[4]

1 Department of Public Health, Federal University of Rio Grande do Norte, Natal, Brazil, 2 Faculty of Health Science of Trairí, Federal University of Rio Grande do Norte, Natal, Brazil, 3 Centro Universitário Santa Maria, Cajazeiras, Paraíba, Brazil, 4 Escola Multicampi de Ciências Médicas do Rio Grande do Norte, Federal University of Rio Grande do Norte, Natal, Brazil

* ketyllemcosta@gmail.com

## Abstract

The objective of this study is to analyze spatial distribution of mortality due to congenital syphilis in Brazil from 2008 to 2022. This is an ecological study that considered congenital syphilis deaths reported in all Brazilian municipalities, from 2008 to 2022, available in the Brazilian government's information systems. We built a thematic map to describe the distribution of congenital syphilis mortality in the country and, subsequently, applied the Local Index Spatial Analysis to identify possible spatial clusters. Finally, we used the Ordinary Least Squares and Geographically Weighted Regression models to identify mortality predictors in the territory. The mortality rate from congenital syphilis was 0.64 deaths per 1,000 live births. The distribution of deaths occurred heterogeneously, with the highest rates in the states of Pará, Acre, Rondônia, Rio de Janeiro and part of Amazonas. We identified statistically significant spatial clusters across the country, with the formation of clusters with a high-high pattern in Pará, Rio de Janeiro, and Mato Grosso (p < 0.05). We observed that the Gini index (p = 0.008; 95% Confidence Interval: 0.02–0.11), the number of nurses in primary care (p = 0.027; 95% Confidence Interval: 0.0005–0.00003) and the proportion of non-treponemal tests by pregnant women (p = 0.016; 95% Confidence Interval: 0.005–0.001) are variables that influence the occurrence of deaths. Congenital syphilis deaths in Brazil occur heterogeneously, with different rates between regions, which are geographically influenced by social and healthcare characteristics of the location.

## Introduction

The number of deaths from congenital syphilis (CS) has increased significantly worldwide in recent years. In 2016, there were 143,000 early fetal deaths or stillbirths and

**Data availability statement:** All relevant data are within the paper and its Supporting information files.

**Funding:** This study was partially funded by the Coordination for the Improvement of Higher Education Personnel- Brazil (CAPES): Financial Code 001. The funders had no role in study design, data collection and analysis, decision to publish, or preparation of the manuscript.

**Competing interests:** The authors have declared that no competing interests exist.

14,000 neonatal deaths due to CS, accounting for more than 40% of all adverse outcomes related to the disease [1]. In 2022, data from the World Health Organization (WHO) estimated 150,000 fetal and stillbirth deaths and 70,000 neonatal deaths [2]. Moreover, the Americas currently face the highest incidence of syphilis globally [3].

The number of deaths from CS in Brazil has increased significantly in recent years [4]. Between 2011 and 2021, the national infant mortality rate due to CS increased by 84.6%, rising from 3.8 to 7.0 deaths per 100,000 live births [5]. Some studies have estimated infant and neonatal mortality due to CS in Brazil, reporting an increase in the mortality rate from 2.74 to 7.84 deaths per 100,000 live births between 2010 and 2017 [6,7].

In Brazil, several strategies have been implemented to reduce the incidence and mortality of congenital syphilis (CS). Ordinance No. 542, of December 22, 1986, established the mandatory reporting of CS, reinforcing its significance as a public health issue [8]. In parallel, the Ministry of Health has strengthened the Family Health Strategy (FHS), implemented in 1994, as a cornerstone of primary health care, with a focus on health promotion and disease prevention [9]. According to data from the 2019 National Health Survey, the FHS had achieved a population coverage rate of 62.6% [10].

Among the initiatives aimed at maternal and child health, the Stork Network stands out as a public policy designed to improve prenatal care, expand access to diagnostic testing for syphilis, and allocate targeted resources for combating the disease within the Unified Health System (SUS) [11]. More recently, the National Pact for the Elimination of Vertical Transmission of HIV, Syphilis, Hepatitis B, and Chagas Disease was established. This initiative sets goals and guidelines aimed at eliminating CS and other transmissible infections in the country by 2030 [12].

In addition, it was observed that the risk of death was higher among children born to untreated mothers or mothers with low educational attainment, in cases with treponemal titers greater than 1:64, and when clinical signs and symptoms were present at birth [3,6,7]. However, despite the currently available literature, there remains a lack of data on the geographic distribution of CS-related deaths in Brazil, as well as on the factors influencing their spatial dynamics. This gap hinders the identification of areas most vulnerable to such outcomes and limits the ability to target control strategies to regions in greatest need.

In this context, analyzing the geographic distribution of CS-related deaths and identifying their influencing factors—based on the most recent data provided by the Brazilian government over a broad historical period (2008–2022)—may yield valuable information to support the identification of areas with a higher propensity for such deaths. This, in turn, can facilitate the development and implementation of more effective preventive interventions tailored to the specific characteristics of each region. Furthermore, the results of such analyses may serve as a reference for other countries with similar conditions to define and implement their own strategies for controlling CS-related deaths.

Given this scenario, an in-depth analysis of the epidemiological situation is necessary to generate evidence that can support the design and implementation of

effective strategies for reducing CS-related mortality in Brazil. Therefore, the objective of this study was to analyze the spatial distribution of congenital syphilis mortality in Brazil from 2008 to 2022.

## Methods

This is an ecological study, developed using secondary data from all municipalities in Brazil. The country comprises 5,570 municipalities, distributed across 27 federative units (26 states and one Federal District). Geographically, it is divided into five macro-regions: North, Northeast, Southeast, South, and Central-West [13]. The study population consisted of new-borns (aged 0–27 days) diagnosed with congenital syphilis in Brazil.

Brazil is the largest country in South America, with a territorial area of 8,509,379.576 km² [14,15] and an estimated population of 212,583,750 [16,17]. Geographically, it is divided into five macro-regions: North, Northeast, Central-West, South, and Southeast (Fig 1) [18]. Additionally, the country has a Human Development Index (HDI) of 0.786 [19] and a primary health care coverage rate of 98.16% [20].

All data were collected in July 2023 from databases linked to the Brazilian Ministry of Health, the Department of Informatics of the Unified Health System (DATASUS), the United Nations Development Programme (UNDP), and the Brazilian Institute of Geography and Statistics (IBGE). The data covered a fifteen-year period, from 2008 to 2022.

For this study, the following data were collected: the number of deaths due to congenital syphilis among newborns aged 0–27 days by municipality of residence at the time of notification, the number of live births, the percentage of non-treponemal tests performed on pregnant women, and the proportion of primary care nurses per inhabitant. The definitions of the variables are described in the variable matrix (Table 1).

In Brazil, congenital syphilis is diagnosed when a newborn is exposed to untreated or inadequately treated maternal syphilis during pregnancy. Diagnosis is based on criteria such as a non-treponemal test titer in the newborn that is ≥ 2 dilutions higher than the maternal titer, the presence of suggestive clinical signs, abnormalities in laboratory tests (including cerebrospinal fluid and radiographic findings), or direct identification of Treponema pallidum. The reactivity of treponemal tests after 18 months of age also confirms infection, due to the persistence of the child's own antibodies. These criteria aim to ensure the early identification of both symptomatic and asymptomatic cases, thereby preventing severe complications [21].

The outcome variable was the average mortality rate due to congenital syphilis (aged 0–27 days), standardized by the population of live births in the year 2015, which represents the median year of the study period. The remaining variables were considered as potential predictors of the outcome variable. The congenital syphilis mortality rate was calculated for each municipality in Brazil using the following equation:

$$Mortality\ Rate = \frac{Mean\ number\ of\ deaths\ from\ congenital\ syphilis\ between\ 2008\ and\ 2022}{Number\ of\ live\ births\ in\ 2015} \times 1{,}000 \tag{1}$$

For the spatial analysis, a thematic map of congenital syphilis (CS) mortality in Brazilian municipalities was initially created. Subsequently, crude mortality rates were smoothed using the local empirical Bayesian method to reduce instabilities caused by municipalities with no reported cases or rates that were markedly different from those of neighboring areas. This procedure produces rates that are more representative of reality, as it accounts for the value of a given municipality and its neighbors, based on a spatial contiguity matrix assigning a value of 1 to municipalities sharing borders and 0 otherwise [22].

Following the descriptive mapping, spatial clusters were identified using Moran's spatial autocorrelation statistics. The Local Moran's Index (Local Indicators of Spatial Association – LISA) was applied to detect and quantify spatial clusters in each municipality, with statistical significance set at $p < 0.05$ [23]. The results were visualized in a Moran Map, showing four spatial patterns: high–high (municipalities with high rates surrounded by high-rate neighbors), low–low, high–low, and low–high.

 

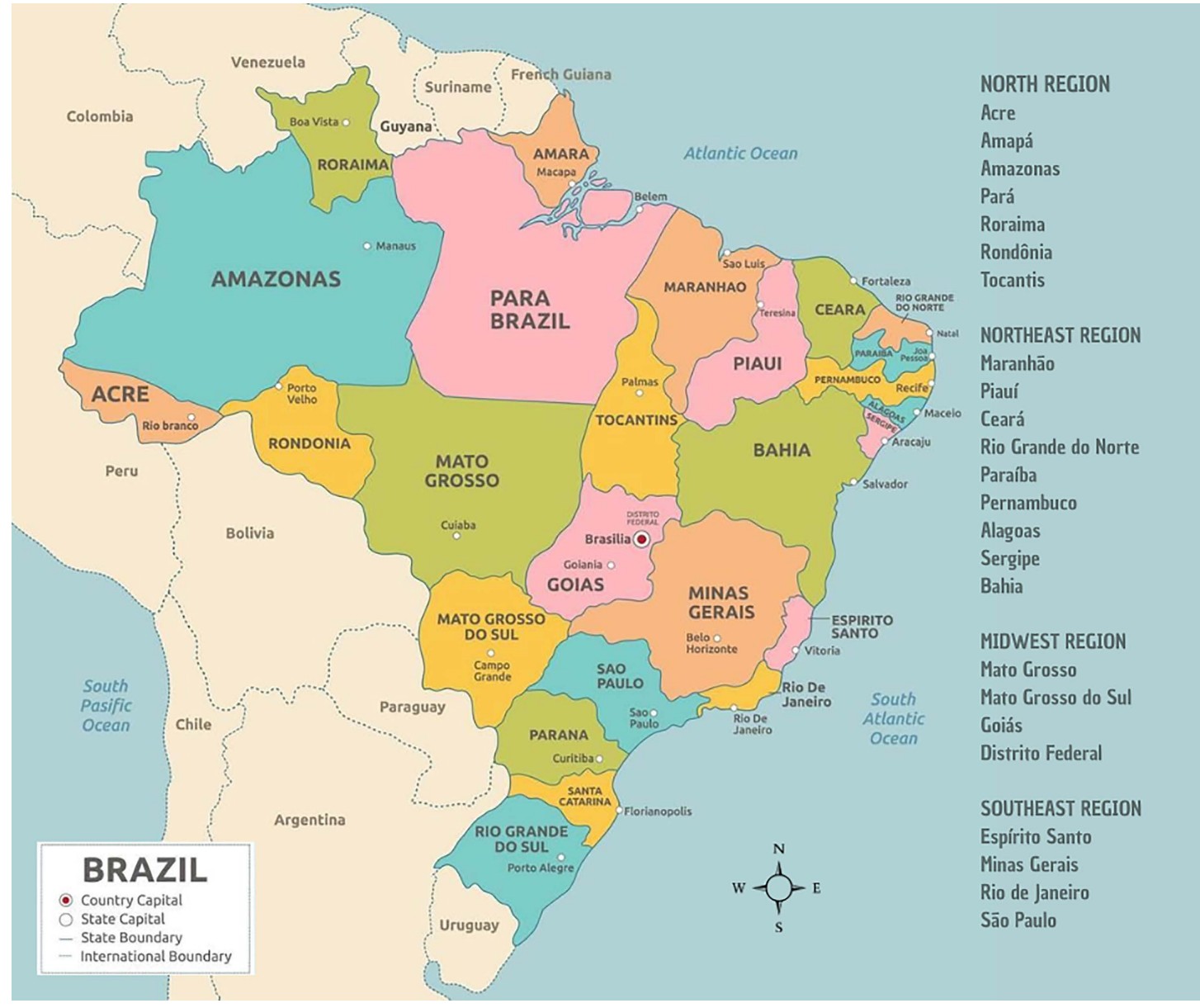

**Fig 1. Geographical division of Brazil. Brazil, 2025.**

The Koenker (Breusch–Pagan) test was applied to assess the non-stationarity of the data. A statistically significant result ($p < 0.001$) confirmed spatial heterogeneity and justified the use of geographically weighted models. Additionally, the Getis–Ord Gi* statistic was employed to detect local hot and cold spots of CS mortality across Brazil. Clusters were defined as contiguous groups of at least three municipalities with Gi* values of the same sign (high–high or low–low). To reduce the risk of false positives, the False Discovery Rate (FDR) correction was applied to Gi* z-scores using the Benjamini–Hochberg procedure, and only clusters remaining significant after adjustment ($q < 0.05$) were retained in the final maps.

**Table 1. Characterization of the dependent and independent variables used in the statistical analysis of the study. Brazil, 2025.**

| Category | Variable | Conceptualization | Source |
|---|---|---|---|
| **Dependent Variable** | | | |
| Mortality | Death rate related to congenital syphilis in newborns aged 0–27 days per reporting residence | Number of deaths from congenital syphilis in newborns aged 0–27 days, divided by the total number of live births in the same place and period, multiplied by 1,000 | Mortality Information System (SIM) – DATASUS |
| **Independent Variables** | | | |
| Demographic | Number of live births | Number of live births reported to the Ministry of Health's Live Birth Information System for each municipality and year | Live Birth Information System (SINASC) – DATASUS/ IBGE |
| Socioeconomic | Gini Index | Measures the degree of income inequality among individuals based on per capita household income; ranges from 0 (perfect equality) to 1 (maximum inequality) | United Nations Development Programme (PNUD)/ IBGE |
| | Municipal Human Development Index (MHDI) | Composite index reflecting longevity, education, and income at the municipal level; ranges from 0 (lowest) to 1 (highest) | PNUD – Atlas of Human Development in Brazil |
| Health Care – Structure | Density of physicians per 10,000 inhabitants | Number of physicians registered in a given location and period, divided by the resident population, multiplied by 10,000 | National Registry of Health Establishments (CNES) – DATASUS |
| | Proportion of primary care nurses per inhabitant | Number of nurses working in primary health care, divided by the population, multiplied by 3,500 (parameter for primary care coverage) | CNES – DATASUS |
| | Primary Health Care (PHC) coverage (%) | Estimated proportion of the population covered by registered primary health care teams | Primary Care Department (DAB) – DATASUS |
| | Percentage of non-treponemal tests per pregnant woman | Number of non-treponemal tests (VDRL or equivalent) performed among pregnant women divided by total pregnant women in the same place and period, multiplied by 100 | Outpatient Information System (SIA) – DATASUS |
| Health Care – Process and Access | Proportion of women initiating prenatal care ≤12 weeks | Percentage of pregnant women who began prenatal care up to the 12th gestational week among total live births | Live Birth Information System (SINASC) – DATASUS |
| | Proportion of women with ≥7 prenatal visits | Percentage of pregnant women who attended seven or more prenatal consultations among total live births | SINASC – DATASUS |
| | Proportion of adequately treated pregnant women | Number of pregnant women diagnosed with syphilis who received adequate treatment according to national guidelines divided by total pregnant women diagnosed | Information System for Notifiable Diseases (SINAN) – DATASUS |

A spatiotemporal scan analysis was also conducted using the SaTScan 9.7 software to identify persistent and emerging clusters over time. The space–time scan statistic used a discrete Poisson model, assuming no geographic overlap of clusters, a maximum cluster size of 50% of the population at risk, circular scanning windows, and 999 Monte Carlo replications. Clusters were reported as statistically significant when $p < 0.05$, and relative risk (RR) values >1 indicated a higher risk of CS mortality compared with the national average [24].

To identify predictors of CS mortality, an Ordinary Least Squares (OLS) regression model was initially fitted using a backward stepwise selection method (removal threshold $p > 0.10$). The OLS model included the following covariates: Gini Index, Municipal Human Development Index (MHDI), density of primary care nurses, proportion of non-treponemal tests performed among pregnant women, physician density, primary health care (PHC) coverage, and prenatal care indicators (early initiation, ≥7 visits, and adequate treatment). Variables were standardized (z-scores), and multicollinearity was assessed using the Variance Inflation Factor (VIF < 5).

Model assumptions (linearity, homoscedasticity, and normality of residuals) were verified prior to spatial diagnostics. The residuals of the OLS model were tested for spatial autocorrelation using the Global Moran's I statistic. Significant

residual autocorrelation (Moran's I = 0.23; $p < 0.001$) confirmed the presence of spatial structure, which was subsequently reduced and became non-significant after applying GWR (Moran's I = 0.04; $p = 0.210$).

The Geographically Weighted Regression (GWR) was then applied to account for spatial non-stationarity and to model spatially varying relationships. The adaptive kernel bandwidth was optimized using cross-validation. To avoid overfitting, only the most significant predictors from the OLS model were included in the GWR (Gini Index, MHDI, nurse density, non-treponemal test proportion, and physician density). Model performance was compared using the Akaike Information Criterion (AIC) for OLS and the corrected Akaike Information Criterion (AICc) for GWR, as recommended by Fothering-ham et al. [25].

The OLS model achieved an $R^2$ of 0.50 (AIC = −8,870), while the GWR model improved the explanatory power to $R^2 = 0.56$ (AICc = −8,880), confirming that accounting for spatial heterogeneity enhanced model fit.

Data tabulation and descriptive analyses were conducted in Microsoft Excel 2016. The local empirical Bayesian rate, Moran's I, and Gi* statistics were calculated in GeoDa 1.20, while the spatiotemporal scan analysis was performed in SaTScan 9.7. The OLS and GWR models were developed in GWR 4.0 and validated using R 4.3.2 (*spdep*, *GWmodel*). All maps were created in QGIS 3.16.

The data used in this study are publicly available, and therefore, ethical approval was not required, in accordance with Resolution No. 466/12 of the Brazilian National Health Council.

## Results

Between 2008 and 2022, a total of 214,203 cases of congenital syphilis were reported in Brazil, of which 1,927 resulted in death due to complications occurring within the first 0–27 days of life, among a population of 2,996,967 live births. Thus, the national mean mortality rate was 0.64 deaths per 1,000 live births. The distribution of mortality was heterogeneous across the country; in most municipalities, the rate ranged between 0.0 and 0.1 per 1,000 live births. However, in some municipalities, it exceeded 1 per 1,000 live births (Fig 2A). The application of the local empirical Bayesian method enabled a clearer visualization of the spatial distribution of this indicator, especially in the states of Pará, Rio de Janeiro, parts of Amazonas, Acre, and Rondônia (Fig 2B).

The Local Moran's Index allowed the identification of spatial clusters throughout the Brazilian territory. High-high clusters were observed in Pará, Rio de Janeiro, Mato Grosso, and, to a lesser extent, in other states (Fig 2C). All clusters showed statistical significance ($p < 0.05$) (Fig 2D).

The Koenker (Breusch–Pagan) statistic confirmed the non-stationarity of the data ($p < 0.001$), supporting the use of geographically weighted models and local spatial statistics. Subsequently, the Getis–Ord Gi analysis identified statistically significant hot and cold spots of congenital syphilis mortality across the country. Hot spots were mainly concentrated in the states of Pará, Rio de Janeiro, and parts of Mato Grosso, while cold spots prevailed in the South and Southeast regions. Clusters were defined as contiguous groups composed of three or more municipalities with significant Gi values of the same sign (high–high or low–low).

To reduce type I error inflation, the False Discovery Rate (FDR) correction was applied to Gi* z-scores using the Benjamini–Hochberg procedure. Only clusters that remained significant after FDR adjustment ($q < 0.05$) were retained in the final mapping (Fig 2E).

In addition, the spatiotemporal risk analysis using the space–time scan statistic (SaTScan, Poisson discrete model) revealed persistent and emerging high-risk clusters between 2008 and 2022. The most likely spatiotemporal cluster was detected in municipalities of Rio de Janeiro, with a 4.7-fold higher risk than the national average ($p < 0.001$), consistent with the results of the purely spatial analysis (Fig 2F).

The application of spatial scan statistics revealed that most areas in the country had a lower risk for this outcome compared to the national average. Moreover, the most likely cluster not to have occurred by chance was identified in

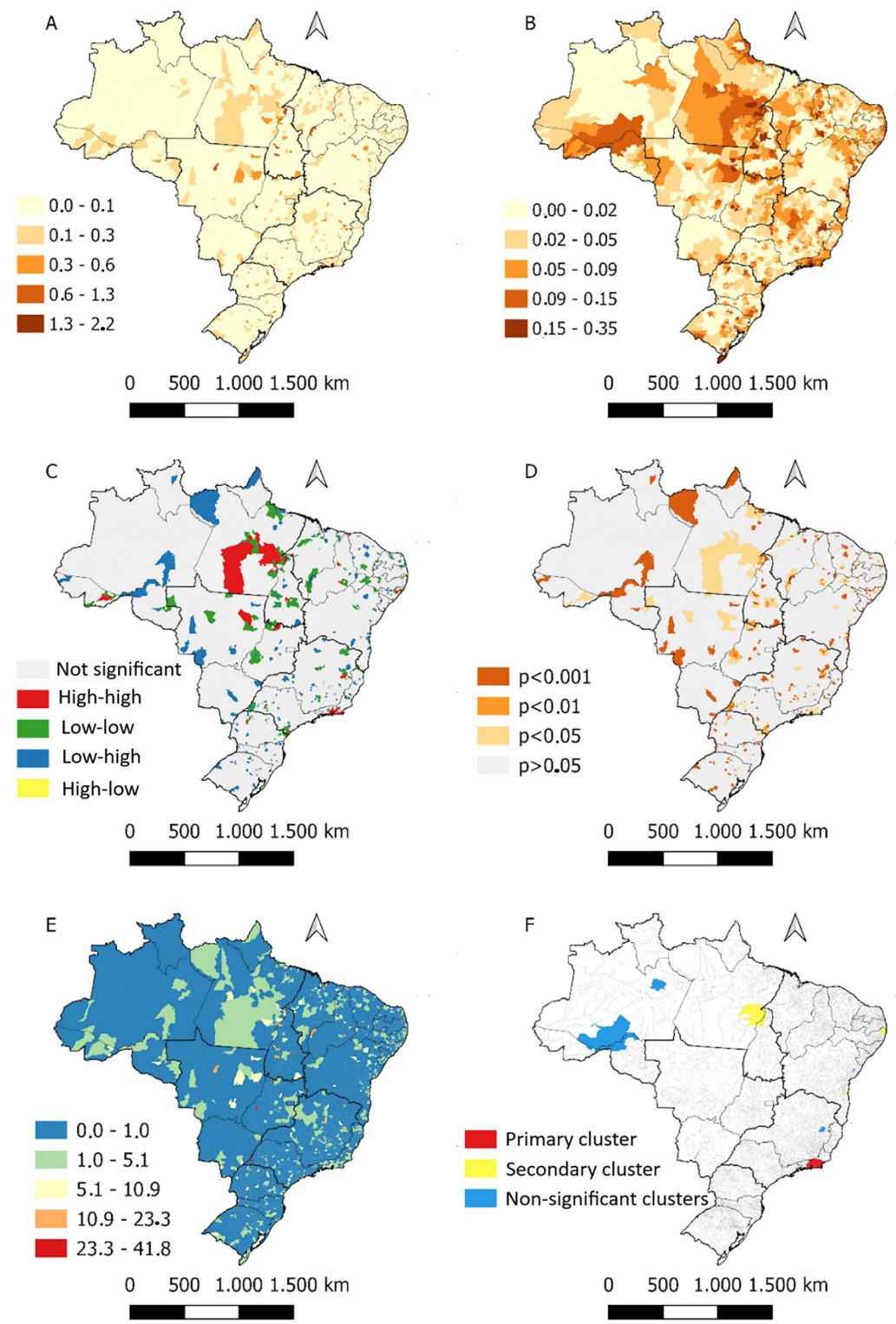

**Fig 2. Spatial pattern of deaths from congenital syphilis in Brazil from 2008 to 2022. Brazil, 2025.**

municipalities in the state of Rio de Janeiro, with a risk 4.7 times higher than the national average, within a radius of 97.8 km (p < 0.001) (Fig 2F).

Based on the ordinary least squares (OLS) regression, higher income inequality (Gini Index) was positively associated with congenital syphilis mortality. Conversely, a greater number of primary care nurses per inhabitant and a higher percentage of non-treponemal tests performed among pregnant women were inversely associated with the outcome. Other variables tested, including physician density, primary health care coverage, and prenatal care indicators, did not remain significant in the final parsimonious model. The OLS model satisfied all statistical assumptions, showing no evidence of multicollinearity (mean VIF = 2.8) and demonstrating a substantial explanatory capacity, with an overall fit of $R^2 = 0.50$ and AIC = −8,870. When spatial heterogeneity was incorporated through geographically weighted regression (GWR), the direction and magnitude of associations remained consistent, but the model fit improved further ($R^2 = 0.56$; AIC corrected = −8,892), confirming that spatially varying relationships provided a more accurate representation of congenital syphilis mortality across Brazilian municipalities (Table 2).

When spatial structure was incorporated through Spatial Lag (SLM) and Spatial Error (SEM) models, both improved the model fit (AIC = −8,875 for SLM; −8,879 for SEM) and showed consistent direction of associations with the GWR. The SLM presented a significant spatial autoregressive parameter ($\rho = 0.31$; p < 0.001), while the SEM showed a significant spatial error coefficient ($\lambda = 0.28$; p < 0.001), confirming residual spatial dependence.

Table 3 presents the regression coefficients (β), standard errors, and p-values for the SLM and SEM models. The results confirm that the Gini index remains positively associated with congenital syphilis mortality, while the density of primary care nurses and the proportion of non-treponemal tests persist as protective factors. The coefficients retained similar magnitudes to those observed in the OLS and GWR models, indicating consistency across the different modeling approaches.

In the geographically weighted model, a positive association was identified between the Gini index and congenital syphilis (CS) mortality, particularly in the states of the Northeast, Southeast, and Central-West regions (Figs 2A and 2B, 3). Conversely, an inverse relationship between the average number of nurses and the outcome was observed throughout the Southeast, South, and Central-West regions (except for Mato Grosso), as well as in the Northeast (except for Piauí and Maranhão) (Fig 2C and 2D). An increase in the availability of non-treponemal tests for pregnant women was associated with a reduction in mortality in the same regions as the previous variable (Fig 2E and 2F).

It is important to highlight that in both models the coefficients were very close to zero and, therefore, should be interpreted with caution. Similarly, the local $R^2$ values were below 10% across the country (Fig 4).

**Table 2. OLS and GWR models for factors associated with congenital syphilis mortality in Brazil, 2008-2022. Brazil, 2025.**

| | OLS Model | | | | GWR Model | |
| --- | --- | --- | --- | --- | --- | --- |
| | β | p-value | CI 95% | | Mean β | Standard Deviation |
| | | | Lower | Upper | | |
| Gini index | 0,06 | 0,008 | 0,02 | 0,11 | 0,06 | 0,005 |
| Municipal Human Development Index (MHDI) | −0.09 | 0.012 | −0.16 | −0.02 | −0.10 | 0.015 |
| Proportion of primary care nurses per inhabitant | −0,0002 | 0,027 | −0,0005 | −0,00003 | −0,0003 | 0,00005 |
| Percentage of non-treponemal tests per pregnant woman | 0,01 | 0,016 | 0,005 | 0,001 | 0,0008 | 0,0001 |
| Physician density per 10,000 inhabitants | −0.015 | 0.044 | −0.029 | −0.001 | −0.013 | 0.004 |
| Primary Health Care coverage (%) | −0.004 | 0.021 | −0.007 | −0.001 | −0.005 | 0.001 |
| Proportion of women initiating prenatal care ≤12 weeks | −0.008 | 0.010 | −0.014 | −0.002 | −0.007 | 0.002 |
| Proportion of women with ≥7 prenatal visits | −0.006 | 0.018 | −0.011 | −0.001 | −0.006 | 0.001 |
| Proportion of adequately treated pregnant women | −0.012 | 0.006 | −0.020 | −0.004 | −0.013 | 0.003 |
| Constant | −0.01 | 0.320 | −0.04 | 0.02 | −0.01 | 0.002 |

**Table 3. Regression coefficients (β), standard errors (SE) and p-values for Spatial Lag Model (SLM) and Spatial Error Model (SEM). Brazil, 2008–2022.**

| Variable | SLM β | SE | p-value | SEM β | SE | p-value |
|---|---|---|---|---|---|---|
| Gini Index | 0.05 | 0.015 | 0.002 | 0.06 | 0.014 | <0.001 |
| Municipal Human Development Index (MHDI) | −0.08 | 0.023 | 0.001 | −0.09 | 0.022 | <0.001 |
| Proportion of primary care nurses per inhabitant | −0.00018 | 0.00006 | 0.004 | −0.00021 | 0.00005 | <0.001 |
| Percentage of non-treponemal tests per pregnant woman | −0.009 | 0.003 | 0.008 | −0.010 | 0.003 | 0.006 |
| Physician density per 10,000 inhabitants | −0.012 | 0.007 | 0.078 | −0.015 | 0.006 | 0.014 |
| Primary Health Care coverage (%) | −0.003 | 0.001 | 0.041 | −0.004 | 0.001 | 0.022 |
| Proportion of women starting prenatal care ≤12 weeks | −0.007 | 0.002 | 0.013 | −0.008 | 0.002 | 0.004 |
| Proportion of women with ≥7 prenatal visits | −0.004 | 0.002 | 0.054 | −0.006 | 0.002 | 0.017 |
| Proportion of adequately treated pregnant women | −0.010 | 0.003 | 0.003 | −0.012 | 0.002 | <0.001 |
| Constant | −0.008 | 0.019 | 0.662 | −0.009 | 0.018 | 0.628 |
| Spatial parameter | ρ = 0.31 (p < 0.001) | – | – | λ = 0.28 (p < 0.001) | – | – |

## Discussion

Syphilis is classified among diseases subject to compulsory notification. The Brazilian Ministry of Health cataloged diseases that pose a risk of spread or dissemination and can generate considerable social impact, creating a notification flow that structures the periodicity and the responsible agency for receiving reports. When a healthcare professional identifies a suspected case, they open a notification form, send the biological sample to the laboratory, and upon receiving the result, confirm the diagnosis and notify health authorities via the Notifiable Diseases Information System (SINAN) [26].

Regarding death certification, only a physician can attest to it, requiring completion of the Death Certificate (DO). In this case, data is entered into the Mortality Information System (SIM). However, studies indicate inconsistencies in congenital syphilis data recorded in both systems, suggesting possible underreporting. It is believed that underreporting is related to poor quality in completing the DO, whether in defining the underlying or associated cause of death, or in the adequate completion of variables [27].

The comparison of the three modeling approaches (OLS, SLM/SEM, and GWR) demonstrated consistent associations across methods, with income inequality remaining a risk factor and primary care nurse density and the proportion of non-treponemal tests acting as protective indicators. The improvement in AIC values in the SLM and SEM models, along with the significance of the spatial parameters (ρ and λ), confirmed that part of the mortality variation is spatially structured. While these spatial models corrected residual dependence, the GWR provided additional refinement by revealing geographic variability in the strength of associations. Together, the results indicate that the predictors are robust across models and that accounting for spatial dependence and non-stationarity is essential for accurately understanding congenital syphilis mortality in Brazil.

Notifications of syphilis and other diseases were indirectly impacted by the COVID-19 pandemic [28,29], since human resources and supplies were directed toward combating the virus. Laboratories prioritized SARS-CoV-2 testing and were unable to meet the demand for other notifiable diseases. Additionally, the population avoided seeking healthcare for non-respiratory conditions, while communication, health education, disease prevention, surveillance, and other mechanisms prioritized COVID-19 [30]. To some extent, this issue negatively impacted prenatal care access due to pregnant women's fear of contracting the virus when attending health services, leading to a reduction in follow-up visits [31].

Our study analyzed the geographic distribution of deaths due to congenital syphilis (CS) among newborns aged 0–27 days in Brazil over a fifteen-year period (2008–2022) and identified associated factors. We observed that most of the country presents a risk of CS-related deaths, particularly in states in the North region (Pará, Acre, Rondônia, and parts of Amazonas) and in Rio de Janeiro. Moreover, mortality is heterogeneously distributed across the Brazilian territory, with

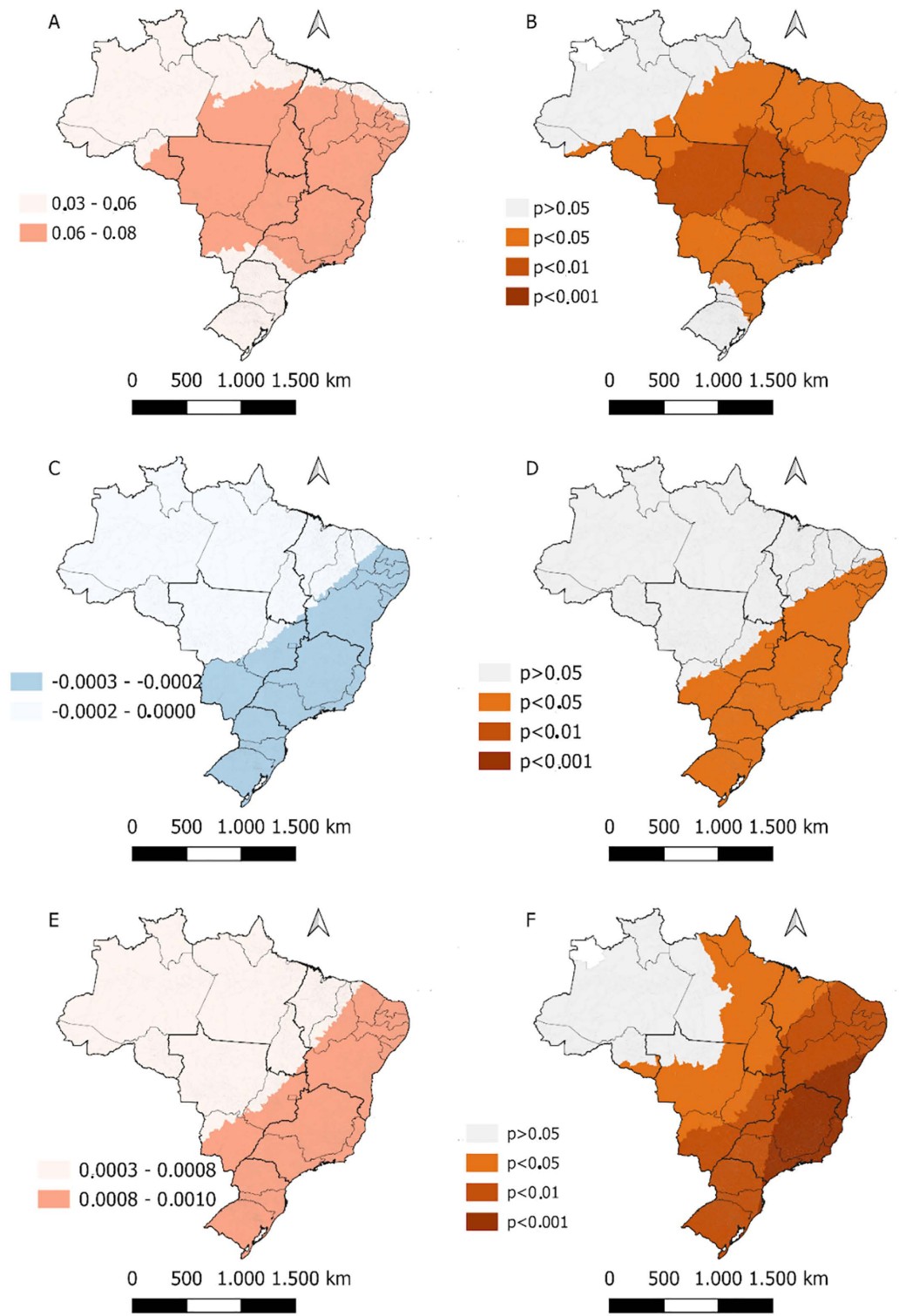

**Fig 3. Coefficient β of local indicators associated with congenital syphilis mortality in Brazil from 2008 to 2022. Brazil, 2025.**

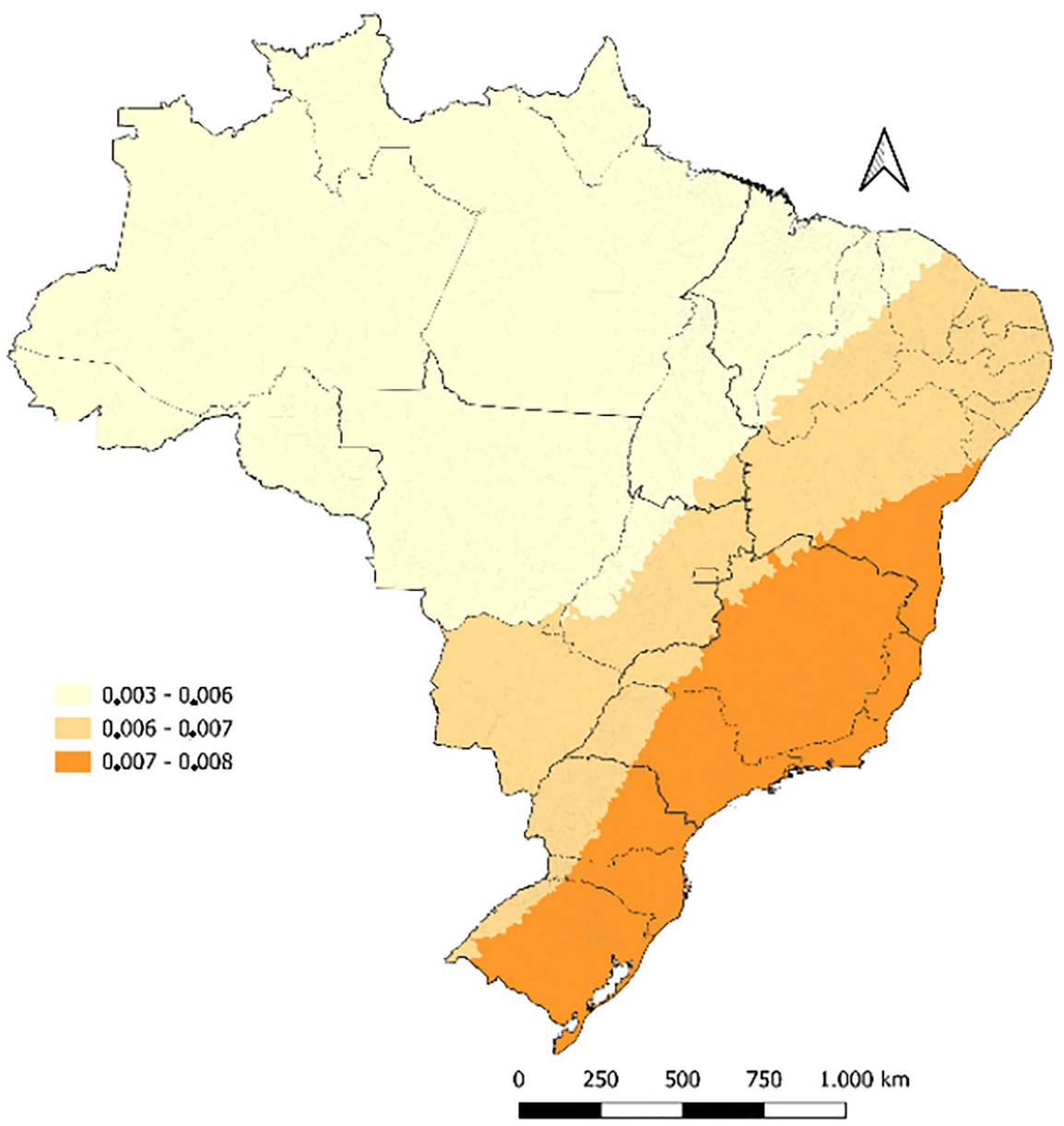

**Fig 4. Local coefficient of determination for indicators (R²) related to congenital syphilis mortality in Brazil from 2008 to 2022. Brazil, 2025.**

the formation of spatial clusters in all states, especially Pará, Rio de Janeiro, and Mato Grosso, which showed a high-high pattern. Finally, we found that the Gini index, the number of nurses in primary care, and the proportion of non-treponemal tests performed on pregnant women are variables that influence the geographic distribution of deaths, particularly in the Northeast, Southeast, Central-West, and South regions.

The robustness of the spatial analysis was reinforced by the application of additional statistical tests. The Koenker (Breusch–Pagan) statistic confirmed the non-stationarity of the data (p < 0.001), supporting the use of geographically weighted regression (GWR) and local spatial statistics. Moreover, the residuals of the OLS model exhibited significant

spatial autocorrelation (Moran's I = 0.23; p < 0.001), which was markedly reduced and became non-significant after the application of GWR (Moran's I = 0.04; p = 0.210), indicating that the local model effectively captured spatial heterogeneity.

In addition, the Getis–Ord Gi analysis identified statistically significant clusters of high and low mortality rates, particularly in Pará, Rio de Janeiro, and Mato Grosso, which remained significant after correction for multiple testing using the False Discovery Rate (FDR) adjustment. The use of the spatiotemporal scan statistic (SaTScan) further strengthened the analysis by revealing persistent and emerging clusters of elevated risk, mainly in the state of Rio de Janeiro. These findings demonstrate that the mortality pattern of congenital syphilis in Brazil is influenced not only by spatial dependence but also by temporal persistence, reinforcing the multifactorial and dynamic nature of this public health issue.

Together, these analytical improvements—supported by a substantial increase in the model's explanatory power ($R^2$ = 0.50 for OLS and 0.56 for GWR) and the correction of model fit metrics (AIC for OLS and AICc for GWR)—confer greater robustness and spatial precision to the results, allowing a more accurate interpretation of the socioeconomic and health determinants of congenital syphilis mortality across Brazilian municipalities.

The Brazilian Ministry of Health reports that both the incidence of CS and related deaths have increased nationwide in recent years [5]. It is estimated that children with CS are twice as likely to die compared to healthy children, which may represent an infant mortality rate (deaths before the age of one year) of 33% [6]. Thus, syphilis remains a significant and ongoing public health issue and one of the main causes of infant mortality, despite being entirely preventable.

In the North region, the largest clusters of CS-related deaths were observed. This is noteworthy given the region's historical context of underdevelopment and socioeconomic and healthcare inequality [32]. Studies indicate that all states in the North region exhibit poverty indicators above the national average [33], as well as considerable disparities in access to healthcare services (with estimates showing that between 1998 and 2003 all Brazilian regions improved access to healthcare, except the North) [34].

Our findings suggest that the higher probability of CS-related deaths in the North may stem from ineffective control of this condition due to inequities in access to services offered by the public health system (SUS). In this region, barriers to accessing healthcare facilities compromise diagnosis, treatment, and follow-up of syphilis cases, thereby increasing the risk of mortality [7]. Furthermore, the low municipal human development index and limited primary healthcare coverage—especially in Pará and Amazonas—combined with insufficient financial resources, significantly hinder the health care system's operation. This often results in healthcare services being concentrated in urban areas, limiting access for rural and riverside populations [35].

The state of Rio de Janeiro also showed a high mortality rate due to CS, despite its relatively high level of socioeconomic development. We hypothesize that inadequate treatment of pregnant women or newborns may have contributed to this result. That is, although Rio de Janeiro offers greater access to diagnostic services and better conditions for disease reporting compared to the North region [36], the persistence of inadequate treatment—or lack of adherence by pregnant women—may worsen the condition and increase the risk of death [37,38]. Additionally, unequal access to health services, especially among Black individuals, adolescents, and those with low educational attainment, may contribute substantially [39]. Therefore, controlling CS and related deaths remains a challenge given the multiple influencing factors.

Previous studies have reported a heterogeneous pattern of CS mortality across Brazil, which can be explained by social (income, education, housing, etc.) and healthcare disparities among states, in addition to the country's vast territorial extension [40,41]. Social conditions—particularly those involving inequities and inequalities—are key risk factors for infant and neonatal mortality, as they affect the allocation of resources for maternal and child healthcare [42]. Thus, a deeper understanding of the problem requires not only an analysis of national public health and social policies, but also state and municipal-level actions, as regional interventions often have greater impact in addressing such conditions [43]. From this perspective, Brazil has made significant progress in public policies aimed at controlling cases and deaths due to syphilis in recent years.

Regarding the inverse association observed between the average number of primary care nurses and the proportion of non-treponemal tests with CS mortality, our findings highlight the need to strengthen primary healthcare. These factors are linked to improved surveillance, and they enable greater preventive and therapeutic assistance for pregnant women and newborns. A systematic review identified the lack of prenatal care as one of the main predictors of neonatal mortality [37].

In addition, it is estimated that appropriate syphilis treatment could reduce neonatal mortality by up to 40% [44]. Therefore, increasing the number of trained nurses in primary care and offering diagnostic testing may reduce barriers to eliminating CS deaths, especially regarding early detection failures and inadequate treatment.

In this study, we also observed that socioeconomic inequality is another key factor predicting CS-related deaths in Brazil. In other words, as inequality—as measured by the Gini index—increases, so do the CS-related deaths. This outcome supports the concept of the social gradient in health proposed by the World Health Organization [45]. Thus, we can affirm that socioeconomic and health conditions vary progressively across territories based on the degree of inequality.

Our findings indicate that CS-related deaths in Brazil demand continued attention from policymakers, health professionals, and researchers. Socioeconomic factors influence mortality both nationally and across macro-regions, suggesting that regional socioeconomic disparities may require targeted actions to control this public health issue [46–48].

The coordination of actions to combat CS at the national level faces several challenges due to Brazil's continental dimensions [49]. Given the disparities across Brazilian regions, tackling CS requires context-specific public health and social policies, tailored to the socioeconomic and healthcare characteristics of each locality [46].

The information presented in our study helps to identify regions where CS is not being effectively controlled, leading to higher risk of deaths from the disease. On a global scale, these findings may serve as a warning for other countries to enhance surveillance in areas with greater social and health vulnerability, thereby enabling more targeted and timely interventions to prevent newborn deaths from CS.

A systematic review conducted in Brazil concluded that the lack of prenatal care was a significant risk factor for congenital syphilis (CS) outcomes. The study demonstrated that newborns are six times more likely to develop CS when their mothers do not receive prenatal care [50]. Thus, early diagnosis and adequate treatment during prenatal care are essential to prevent CS [51]. A study conducted in China found that newborns of women diagnosed with syphilis after 36 weeks of gestation had 25 times higher odds of developing CS [52]. Therefore, diagnosis and treatment in the third trimester of pregnancy carry a greater risk for adverse outcomes [53].

Late initiation of prenatal care is a prominent issue in developing countries. Studies conducted in sub-Saharan African countries, such as Ethiopia, Tanzania, and South Africa, demonstrate that this situation is prevalent. Moreover, it may be aggravated by socioeconomic and demographic factors, such as the concentration of health services in urban areas, high transportation costs hindering access, and low educational levels. As evidenced in the literature, this scenario is similar to that found in Brazil [54–57].

In contrast to syphilis, vertical transmission of HIV, a virus also tested during prenatal care, is decreasing in Brazil. Nevertheless, the country has requested certification of elimination of vertical HIV transmission from the Pan American Health Organization (PAHO/WHO). This divergent scenario between the diseases may be influenced by two main factors: reinfection of the pregnant woman by an untreated partner and risk compensation associated with the use of pre-exposure prophylaxis (PrEP) [58].

Syphilis can recur in pregnant women due to untreated partners, making reinfection possible even after successful treatment. Therefore, partner treatment is strongly recommended. Risk compensation refers to the possibility of increased risky behaviors due to a sense of protection, such as that provided by PrEP [59]. In this case, risky behaviors may be primarily associated with inconsistent condom use and an increase in the number of sexual partners [60]. To address this issue, the importance of health education is emphasized, and PrEP should not be undervalued given its significant impact on reducing HIV in Brazil and worldwide [61].

A limitation of this study lies in the use of secondary data provided by the Brazilian government and the possibility of underreporting. However, we did not observe significant changes in the data collection or reporting structure that would materially affect the results. However, studies indicate that a significant number of cases remain underreported, meaning they are either undiagnosed or not recorded by healthcare professionals.

Furthermore, there is a limitation related to the small number of independent variables included in the geographically weighted regression (GWR) analysis. This methodological choice was made to ensure the stability of local coefficients and to minimize the effects of multicollinearity among predictors. Although this approach allowed for a more parsimonious interpretation of the results, it is possible that other relevant variables were excluded from the final model.

Additionally, it is recognized that the restriction on the number of variables may limit the explanatory capacity of the model in certain spatial contexts. Therefore, it is recommended that future research explore a broader set of determinants, including contextual, social, and structural factors, in order to enhance the understanding of spatial patterns of congenital syphilis mortality in Brazil.

## Conclusion

There is a heterogeneous distribution of CS-related deaths in Brazil, characterized by a risk of occurrence in both less developed and more developed regions, as well as the formation of high-high spatial clusters in several states. The distribution dynamics of mortality must be analyzed in light of socioeconomic and healthcare variables, such as the Gini index, the proportion of nurses in primary care per inhabitants, and the proportion of non-treponemal tests performed in pregnant women, as these are key factors directly influencing mortality patterns across the territory.

Finally, we recommend that further studies explore the reasons behind CS-related deaths in regions with better socioeconomic conditions, as these areas would ideally exhibit lower morbidity and mortality due to better access to healthcare services.

## Supporting information

**S1 File. Database of congenital syphilis in Brazil.**
(XLSX)

## Author contributions

**Conceptualization:** Yago Tavares Pinheiro, Richardson Augusto Rosendo da Silva, José Rebberty Rodrigo Holanda.

**Data curation:** Yago Tavares Pinheiro, Richardson Augusto Rosendo da Silva, Cristiane da Silva Ramos Marinho, José Rebberty Rodrigo Holanda.

**Formal analysis:** Yago Tavares Pinheiro, Janmilli Dantas da Costa, Cristiane da Silva Ramos Marinho, Jurandir Alves de Freitas Filho, Luennia Kerlly Alves Rocha, Victória Sampaio Moreira, Ruan Carlos de Queiroz Monteiro.

**Funding acquisition:** Angelo Giuseppe Roncalli da Costa Oliveira.

**Investigation:** Yago Tavares Pinheiro, Janmilli Dantas da Costa.

**Methodology:** Yago Tavares Pinheiro, Angelo Giuseppe Roncalli da Costa Oliveira, Janmilli Dantas da Costa, Cristiane da Silva Ramos Marinho.

**Project administration:** Richardson Augusto Rosendo da Silva, Angelo Giuseppe Roncalli da Costa Oliveira, José Rebberty Rodrigo Holanda.

**Resources:** Yago Tavares Pinheiro.

**Software:** Yago Tavares Pinheiro.

**Supervision:** Richardson Augusto Rosendo da Silva, Angelo Giuseppe Roncalli da Costa Oliveira, José Rebberty Rodrigo Holanda.

**Writing – original draft:** Yago Tavares Pinheiro, Janmilli Dantas da Costa, Cristiane da Silva Ramos Marinho, Jurandir Alves de Freitas Filho, Luennia Kerlly Alves Rocha, Victória Sampaio Moreira, Ruan Carlos de Queiroz Monteiro.

**Writing – review & editing:** Richardson Augusto Rosendo da Silva, Ketyllem Tayanne da Silva Costa, Angelo Giuseppe Roncalli da Costa Oliveira, José Rebberty Rodrigo Holanda.

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
