## [Decision Letter · Decision Letter 0]

15 Jul 2025

Dear Dr. Costa,

**Please address all reviewer comments, and make sure to implement the analyses suggested by Reviewer #2, along with a more thorough evaluation of the model assumptions (including the possible use of alternative models) and the inclusion of additional variables that may be relevant to the context studied**

We look forward to receiving your revised manuscript.

Kind regards,

Vinícius Silva Belo

Academic Editor

PLOS ONE

2. We note that Figures 1 A-F, 2 A-F, 3 in your submission contain [map/satellite] images which may be copyrighted. All PLOS content is published under the Creative Commons Attribution License (CC BY 4.0), which means that the manuscript, images, and Supporting Information files will be freely available online, and any third party is permitted to access, download, copy, distribute, and use these materials in any way, even commercially, with proper attribution. For these reasons, we cannot publish previously copyrighted maps or satellite images created using proprietary data, such as Google software (Google Maps, Street View, and Earth). For more information, see our copyright guidelines: http://journals.plos.org/plosone/s/licenses-and-copyright.

1. You may seek permission from the original copyright holder of Figures 1 A-F, 2 A-F, 3 to publish the content specifically under the CC BY 4.0 license.  

Additional Editor Comments (if provided):

Reviewers' comments:

Reviewer's Responses to Questions

**Comments to the Author**

1. Is the manuscript technically sound, and do the data support the conclusions?

Reviewer #1: Yes

Reviewer #2: No

2. Has the statistical analysis been performed appropriately and rigorously?

Reviewer #1: Yes

Reviewer #2: No

3. Have the authors made all data underlying the findings in their manuscript fully available?

Reviewer #1: No

Reviewer #2: Yes

4. Is the manuscript presented in an intelligible fashion and written in standard English?

Reviewer #1: Yes

Reviewer #2: Yes

Reviewer #1: The manuscript “SPATIAL ANALYSIS OF MORTALITY DUE TO CONGENITAL SYPHILIS IN BRAZIL FROM 2008 TO 2022” develops a spatial analysis on congenital syphilis in the period from 2008 to 2022.

The topic of congenital syphilis is of extremely high relevance, especially because syphilis is a public health problem that has gained worldwide attention due to the increase in cases in countries considered high-income - a very old problem, but one that has resurfaced.

It is observed that the manuscript is well written and the methodology is well documented, therefore the recommendations will only be to improve it a little more before it is accepted for publication.

[1] Introduction

The introduction to the article is very good, quite concise and objective, and the authors also manage to make the objectives of the research clear in the introduction. However, the authors could better contextualize the scenario of syphilis in Brazil.

Brazil is currently one of the countries with a significant number of scientific publications on this topic. It is known that the country declared a syphilis epidemic in 2016 and that important actions were implemented after that, such as the “No Syphilis” Project and the Seals for the Elimination of Vertical Transmission of Syphilis and HIV. As far as the international scene is concerned, this includes the World Health Organization (WHO), Brazil even submitted the report on the elimination of vertical transmission of HIV.

I strongly recommend that the authors delve a little deeper into the history of syphilis in Brazil. It is important, because the country has made significant progress in this area.

Another important, but not mandatory, point is that authors can present in the introduction an overview of the epidemiological scenario regarding syphilis in the world. To do this, they can include data from the United States, Canada, China and countries in the European Union. This type of information is important because it shows that syphilis is not a problem only in developing or low- and middle-income countries.

[2] Methodology

The methodology is adequate, congratulations, but the authors need to make all the data available, although the authors state that all the files are available in the DATASUS database, there is a difficulty in obtaining the data, an aspect that hinders the transparency and reproducibility of the experiment - I tried to download it and it really is not easy at all. This link https://datasus.saude.gov.br/informacoes-de-saude-tabnet/ points to several possible paths that do not lead to the data set used by the authors, sometimes it is also not accessible - I tested it myself.

The data can be made available in a file with the extension “.csv” as a supplementary file to the manuscript or made available in a public domain repository, such as Zenodo, which also has a DOI. Authors must not only make available all the data used, they must create a data dictionary describing each field used.

Although the authors report that the data are available in the DATASUS database, it appears that there are data that are not in this scope, for example, the Unified Health System of the United Nations Development Program (UNDP) and the Brazilian Institute of Geography and Statistics (IBGE) are not available in DATASUS. For this reason, all data must be available as recommended - they must be easily accessible to everyone, together with the manuscript when published.

[3] Discussions

The discussions conducted by the authors are rich and very useful - congratulations. However, the limitations of the study are not addressed. Brazil is a country that presents several issues regarding syphilis, especially regarding the reporting of cases of congenital syphilis, therefore it is necessary to consider these issues, as they present a major limitation for epidemiological studies.

Despite having submitted a report on the elimination of vertical transmission of HIV to the World Health Organization (WHO), Brazil has indicators of congenital syphilis that point in the opposite direction. How is this possible if women notified of HIV during pregnancy are subject to the same health system as women notified of syphilis? Brazil has a history of purchasing dual tests for syphilis and HIV that exceed 80 million dollars per year, in other words, there is no shortage of tests in Brazil.

The indicators of congenital syphilis in Brazil are the same or worse when compared to countries in the sub-Saharan region of Africa. How is this possible? We know that Brazil has a much more developed primary health care system than countries in the sub-Saharan region of Africa. Another point that draws attention is that countries in the sub-Saharan region of Africa maintain poor indicators for HIV (vertical transmission) and for Syphilis (vertical transmission), and not only for syphilis, as seems to be the case in Brazil.

In this context, I recommend that the authors delve deeper into this discussion. Is there not a problem in the definition of congenital syphilis cases in Brazil? Does Brazil actually investigate cases of congenital syphilis? Regarding notification of the cause of death, which Health Information System does Brazil use for this? Is it an easy-to-use system? Does it really ensure that deaths are recorded appropriately? In Brazil, I remember a great debate about the notification of deaths during the COVID-19 pandemic, precisely because health professionals had difficulty using ICD-10 in the death notification system.

There are publications of articles by Brazilian researchers reporting problems in reporting cases of congenital syphilis. Brazil also appears to have published a report that addresses this same issue - more than 90% of reported cases of congenital syphilis are based on doubt. Why does this happen in Brazil? Does Brazil have a system for monitoring cases of syphilis in pregnant women that allows public health authorities to assess these cases in real time (online) and monitor follow-up?

These are issues that can be addressed in the discussion - given the quality of the manuscript (which is very good) it is important that the authors really delve into these issues.

Congratulations to the authors and thank you for the opportunity to review this excellent work.

Reviewer #2: Dear authors,

Although the importance of the addressed subject, I made some suggestions to improve the quality of results and, consequently, of the manuscript. Please, see the suggestions bellow:

Abstract:

It should be improved mentioned the originality of the study.

Introduction:

The authors should evidenced that the studied problem is not only restricted to Brazil. Nothing was told about the epidemiological scenery worldwide;

Quite confuse these lines "Between 2011 and 2021, there was a 39.9% rise in deaths from the disease

nationwide. Moreover, during the same period, the infant mortality rate due to CS increased by

84.6%, rising from 3.8 to 7.0 deaths per 100,000 live births [2]:"

In the introduction sounds that Brazil doesn't have any policy to fight the addressed problem. I suggest the authors talked a little bit about it;

What was the reason for choosing the studied period (2008-2022)? In the sentence "based on the most recent data provided by the Brazilian

government over a broad historical period (2008 to 2022)—" sounded you had a reason for that;

Methods:

I suggest the authors explained a little bit more about the study scenery. It is very well known the social inequity among the Brazilian regions. What about prenatal coverage? What about medical coverage? What about geography discrepancies? .....

There should be a map of Brazil showing the states and region limits to facilitate the reading and comprehension;

Nothing was told about the criteria for congenital syphilis (CS) diagnosis. Did you considered it when collected the data? This should be mentioned since you are working with the CS mortality rate;

Why did you consider such few independent variables? For example, you only focused on primary nurse and not medical doctors. CS is great influenced by health coverage, schooling level, the gestational age of the first antenatal care, socioeconomic conditions, number of prenatal consults, adequate treatment, .... I suggest you to expand your variables;

In autocorrelation analysis, considering the non-stationary of the data (please, apply statistic test to comprove it!), I suggest the authors to also apply GI statistics. Remember to define what you mean by group. For example, some studies consider a cluster only those composed by 3 or more municipalities. Additionally, you should apply False Discovery Ratio (FDR) test to exclude the non-significant clusters;

Additonally, considering the temporal subject, I suggest the authors to employ the spatiotemporal risk analysis and not only focus on spatial risk;

Again, GWR was very restricted to few idependent variables. Furthermote, nothing was told about the spatial dependency of OLS and GWR residuals. Just remembering that AIC is for OLS and AICc for GWR. What was the method you applied in OLS? Stepwise?

Results:

I bet the results will change a lot after data reanalysis. Therefore, I will only evaluate in the second version. However, I suggest you to construct the figures separately for each topic. For example, Fig 1 should show the spatial distribution and autocorrelation; Fig 2 spatiotemporal and spatial risk analysisl, Fig. 3 GWR;

- Title of figures should express the intention of them. For example, Fig 2 show the beta coefficients? and 3 the local r2? Although we referred the figs in the manuscript, the titles were quite vagues.

Be careful when applying "," in numbers . In English we use "." ;

Discussion:

Discussion was focused exclusively in SDH, while policies were discarded. For example, the authors considered the low access to primary healthcare in North associated to CS mortality while in Rio is offered greater access. Previous studies in Rio de Janeiro showed that it is not true. The great problem in fighting CS in Rio is basically the unequal access mainly in women living in slams. The high-high cluster in North is composed by rich municipalities in Pará and they attracted immigrant searching for better life conditions. Also, the ones in Southeast have the greater primary healthcare coverage. Consider evaluate previous publications exploring the CS problem. The cluster in Rio is basically composed by the metropolitan municipalities. A super inequity region;

Please, consider taking into account the efficiency of polices fighting CS in the address problem;

Discussion and data quality will increase after reanalyzing data;

The study limit is already know for the ecologic study. I suggest to exclude it;

This sentence "Finally, we recommend that further studies explore the reasons behind CS-related deaths in

regions with better socioeconomic conditions, as these areas would ideally exhibit lower

morbidity and mortality due to better access to healthcare services", should be moved to conclusion;

Additionally, conclusion should answer all the goals of the paper. Nothing was told about the GWR results,....

**Do you want your identity to be public for this peer review?** For information about this choice, including consent withdrawal, please see our Privacy Policy

Reviewer #1: **Yes:** Ricardo Valentim

Reviewer #2: No

---

## [Author Response · Author response to Decision Letter 1]

4 Sep 2025

Dear Editors and Reviewers,

We would like to express our sincere gratitude for the careful review and thoughtful comments provided on our manuscript entitled “Spatial analysis of mortality due to congenital syphilis in Brazil from 2008 to 2022”. The suggestions and observations have been thoroughly considered and have significantly contributed to the improvement of the manuscript. Below, we present our point-by-point responses, outlining the changes made to the text and providing justifications where necessary. Should there be any remaining questions or a need for further modifications beyond those already made, we remain at your disposal.

---

## [Decision Letter · Decision Letter 1]

29 Sep 2025

Dear Dr. Costa,

We look forward to receiving your revised manuscript.

Kind regards,

Vinícius Silva Belo

Academic Editor

PLOS ONE

Journal Requirements:

**Additional Editor Comments:**

After carefully considering the recommendations made by Reviewer 2, who is no longer able to reassess the manuscript, I have conducted a detailed evaluation of the article.

It is necessary that the analyses related to the predictor variables be revised. The justification provided for not including relevant variables (“A considerable number of studies have already examined the variables mentioned… Therefore, we decided to include variables that we consider important but that have been scarcely addressed in previous studies. This constitutes one of the novel aspects and strengths of the present study”) is inadequate. Even if certain variables have already been studied, they should still be included in the analyses to ensure proper control for confounding, to allow collinearity assessment, and to strengthen the contribution of your results to the literature. The very low R² values obtained in your models reinforce the need to include a broader set of determinants.

In addition, given the spatial nature of the data and the presence of spatial dependence (as your own LISA results demonstrated), the OLS model is unlikely to be the most appropriate. Therefore, the data should be reanalyzed with a broader set of explanatory variables and also by considering additional models, such as the Spatial Lag Model (SLM) and the Spatial Error Model (SEM), in addition to the GWR.

Reviewers' comments:

Reviewer's Responses to Questions

**Comments to the Author**

Reviewer #1: All comments have been addressed

2. Is the manuscript technically sound, and do the data support the conclusions?

Reviewer #1: Yes

3. Has the statistical analysis been performed appropriately and rigorously?

Reviewer #1: Yes

4. Have the authors made all data underlying the findings in their manuscript fully available?

Reviewer #1: Yes

5. Is the manuscript presented in an intelligible fashion and written in standard English?

Reviewer #1: Yes

Reviewer #1: The authors strive to meet all recommendations. I consider the manuscript to be of good quality, so I am in favor of its approval.

**Do you want your identity to be public for this peer review?** For information about this choice, including consent withdrawal, please see our Privacy Policy

Reviewer #1: **Yes:** Ricardo Valentim

---

## [Author Response · Author response to Decision Letter 2]

31 Oct 2025

Dear,

All requested adjustments have been accepted. We appreciate all suggestions for improvement.

---

## [Editor Report · Decision Letter 2]

10 Nov 2025

Dear Dr. Costa,

Thank you for submitting your manuscript to PLOS ONE. After careful consideration, we feel that it has merit but does not fully meet PLOS ONE’s publication criteria as it currently stands. Therefore, we invite you to submit a revised version of the manuscript that addresses the points raised during the review process.

**While you have made clear improvements in the manuscript, including the incorporation of the SLM and SEM models, only the AIC values and spatial parameters (ρ and λ) are presented, while the regression coefficients for the explanatory variables in these models are missing. Please include a concise table reporting the coefficients (β) and p-values for the SLM and SEM models. This will ensure transparency and allow readers to confirm whether the associations remain consistent across models. The Discussion should also better articulate and compare the results obtained from the three modeling approaches (OLS, SLM/SEM, and GWR).**

We look forward to receiving your revised manuscript.

Kind regards,

Vinícius Silva Belo

Academic Editor

PLOS ONE
---

## [Author Response · Author response to Decision Letter 3]

17 Dec 2025

Reviewer

Comment 1: “While you have made clear improvements in the manuscript, including the incorporation of the SLM and SEM models, only the AIC values and spatial parameters (ρ and λ) are presented, while the regression coefficients for the explanatory variables in these models are missing. Please include a concise table reporting the coefficients (β) and p-values for the SLM and SEM models. This will ensure transparency and allow readers to confirm whether the associations remain consistent across models. The Discussion should also better articulate and compare the results obtained from the three modeling approaches (OLS, SLM/SEM, and GWR).”

Response: The reviewer's suggestion was accepted, and the information was added to Table 3 (Results section). Additionally, a brief discussion of the new results was included in the third paragraph of the Discussion section.

Sincerely,

Yago Tavares Pinheiro, PhD

---

## [Editor Report · Decision Letter 3]

21 Dec 2025

SPATIAL ANALYSIS OF MORTALITY DUE TO CONGENITAL SYPHILIS IN BRAZIL FROM 2008 TO 2022

PONE-D-25-29476R3

Dear Dr. Costa,

We’re pleased to inform you that your manuscript has been judged scientifically suitable for publication and will be formally accepted for publication once it meets all outstanding technical requirements.

Kind regards,

Vinícius Silva Belo

Academic Editor

PLOS One
---

## [Editor Report · Acceptance letter]

PONE-D-25-29476R3

PLOS One

Dear Dr. Costa,

I'm pleased to inform you that your manuscript has been deemed suitable for publication in PLOS One. Congratulations! Your manuscript is now being handed over to our production team.

Kind regards,

on behalf of

Dr. Vinícius Silva Belo

Academic Editor

PLOS One